# An interpretable deep learning framework for predictive modeling of postoperative infections in ICU patients

Xiaoyu Wu[1], Isaac Luria[2], Meisheng Xiao[1], Patrick Tighe[3], Fei Zou[1,4], Baiming Zou[1,5*]

1 Department of Biostatistics, University of North Carolina at Chapel Hill, Chapel Hill, North Carolina, United States of America, 2 Department of Anesthesiology, University of Florida, Gainesville, Florida, United States of America, 3 Department of Orthopedic Surgery, University of Florida, Gainesville, Florida, United States of America, 4 Department of Genetics, University of North Carolina at Chapel Hill, Chapel Hill, North Carolina, United States of America, 5 School of Nursing, University of North Carolina at Chapel Hill, Chapel Hill, North Carolina, United States of America

* bzou@email.unc.edu

## Abstract

A significant proportion of intensive care unit (ICU) patients undergo surgical procedures, and some may develop postoperative infections. Accurately predicting postoperative infection risk and identifying key contributing factors is crucial for improving postoperative management and understanding infection mechanisms. However, this task is challenging due to the complex interplay of multiple risk factors. While machine learning models can model these intricate associations to predict postoperative infection risk, their lack of interpretability – failing to uncover each factor's impact— hinders their adoption in clinical settings. To address this difficulty, we introduced an interpretable deep neural network (DNN) model that integrates a permutation feature importance test (PermFIT). PermFIT rigorously evaluates the impact of each feature on postoperative infection risk through a rigorous statistical inference. By using only the identified important features as inputs, the DNN's predictive performance can be further enhanced. We conducted an extensive study using electronic health records (EHRs) from the Medical Information Mart for Intensive Care (MIMIC-III), a large-scale ICU EHR database. Under the PermFIT framework, our DNN model effectively identifies significant factors associated with postoperative infections while delivering the most accurate postoperative infection risk predictions. These findings highlight the clinical utility of our proposed DNN framework in managing postoperative care for ICU surgical patients, ultimately improving their health outcomes.

## 1 Introduction

Millions of patients admitted to intensive care units (ICUs) may need to undergo different types of surgical procedures, facing a heightened risk of developing postoperative infections [1]. The postoperative infections have a prevalence rate ranging

**Data availability statement:** The analytic R codes to duplicate the analysis results are available on GitHub (https://github.com/BZou-lab/DeepInfection). The data set used in the analysis is publicly available and extracted from the MIMIC-III database at https://physionet.org/content/mimiciii/1.4/.

**Funding:** This study was partially supported by NIH (National Institutes of Health) R56 (1R56LM013784) and R01 (R01LM014407 and 1R01HL173044) grants.

**Competing interests:** The authors have declared that no competing interests exist.

from 3.0% to 20.7% and an incidence rate of 5–10% in tertiary care hospitals, with rates rising to 28% among ICU patients [2,3]. The public health impacts of postoperative infections are severe, leading to prolonged hospital stays, increased healthcare costs, and higher morbidity and mortality rates [4–6]. Clinically, postoperative infections can cause a range of complications, including sepsis, organ dysfunction, and delayed wound healing, which further complicate patient recovery [6,7]. Additionally, postoperative infections are significantly associated with increased morbidity and mortality for ICU surgical patients [8]. For instance, in 2015, approximately 687,000 hospital-acquired postoperative infections were reported in U.S. acute care hospitals, resulting in around 72,000 deaths [6]. Understanding and mitigating the risk of postoperative infections for ICU surgical patients is crucial for improving this patient population's health outcomes and reducing the burden on healthcare systems [6].

Preventive measures such as proper preoperative screening, antimicrobial prophylaxis, and stringent infection control protocols are crucial in mitigating the impact of postoperative infections on ICU patients [9,10]. Accurate prediction of postoperative infection risk and identification of important risk factors are paramount for implementing timely and effective preventive measures. By gaining an in-depth understanding of specific factors associated with postoperative infection risk, healthcare providers can tailor interventions to mitigate these risks and optimize patient outcomes. Early identification of patients at higher risk allows for developing targeted interventions and close monitoring, potentially reducing the overall incidence of infections and minimizing associated morbidity and mortality [11]. Therefore, a comprehensive understanding of postoperative infection factors is essential in guiding clinical decision-making and improving ICU patient care in surgical settings.

However, achieving these objectives could be difficult since it would require constructing an effective model for accurately predicting postoperative infection risk and identifying important associated factors. This can be challenging due to the complex interplay among various risk factors and outcomes. Factors such as patient demographics, comorbidities, surgery type, and procedure-specific variables all impact the onset and severity of postoperative infections [12–14]. Additionally, the dynamic nature of patient health and the evolving landscape of healthcare-associate infections further complicate predictive modeling efforts [15]. Despite these challenges, developing robust predictive models holds immense potential for informing clinical decision-making, optimizing resource allocation, and reducing the burden of postoperative infections on both patients and healthcare systems. Therefore, advancing predictive modeling techniques is essential for improving surgical care and patient safety for ICU patients.

Most existing predictive models for postoperative infection risks employ parametric methods with restrictive assumptions [16,17], which can not be verified and often fail to hold in real-world scenarios [18]. In contrast, machine learning methods have been employed to construct predictive models, dealing with complex associations between risk factors and outcomes [19–21]. This study investigates and compares commonly used machine learning methods: Support Vector Machine (SVM) [22,23], eXtreme Gradient Boosting (XGBoost) [24,25], Random Forest (RF) [26,27], and Deep

Neural Network (DNN) [28,29] regarding their performance in predicting postoperative infection risk among ICU surgical patients. These methods leverage the complex associations between input predictors and postoperative infection risk without requiring the specification of functional forms between predictors and outcomes, as traditional statistical models like logistic regression do [30]. Though the DNN method can approximate complex associations, the conventional DNN method can be unstable under finite sample size settings due to random parameter initialization, leading to poor prediction accuracy. To address this issue, this study adopts a stable DNN ensemble as proposed in our earlier study by employing a novel bootstrapping and filtering procedure to exclude poorly performing DNN models, with the final prediction based on the mean of the top-performing models [31].

While machine learning models provide robustness and flexibility, they are black-box models lacking transparency in interpreting the impact of each individual feature on postoperative infection risk [32–35], hindering their adoption in clinical practice. To address this difficulty, we introduce a novel, computationally efficient, and interpretable feature importance identification tool for machine learning models, the permutation-based feature importance test (PermFIT) framework [36]. This framework allows for a rigorous evaluation of the impact of each individual feature input to each machine learning model. A notable difference between our feature importance identification framework and existing machine learning methods for postoperative infection prediction is that our framework not only enables flexible modeling of complex associations but also facilitates robust identification of significant features with rigorous statistical inference on feature importance scores. Existing machine learning methods only provide a relative ranking of feature importance scores (e.g., SHAP and LIME), potentially rendering the identified important features statistically insignificant [37,38]. Identifying the important features not only enhances our understanding of postoperative infection mechanisms but also potentially improves model prediction performance [39,40]. To the best of our knowledge, this is the first work to develop an inherently interpretable framework for postoperative infection prediction with the capability to evaluate the impact of each individual potential risk factor. We demonstrate the clinical utility of our framework in a prognostic analysis, wherein the stable DNN model achieved the best prediction performance for all evaluation metrics among all machine learning models considered, based on real-world large-scale ICU patients' electronic health records (EHRs).

## 2  Methods

**EHRs for ICU Patient Postoperative Infection Risk Predictions** In this study, we utilized data extracted from the Medical Information Mart for Intensive Care (MIMIC-III) database [41], which includes EHRs of patients admitted to the ICUs of Beth Israel Deaconess Medical Center from 2001 to 2012, encompassing over 40,000 individuals. Our analysis began with 20,065 admission records involving surgical procedures. By retaining the most recent admission record for patients with multiple surgical admissions, we obtained EHRs for 17,611 unique surgical patients. After excluding records with missing values, our dataset included 13,718 patients (Fig 1A). Postoperative infection information was derived based on International Classification of Diseases (ICD-9) codes [42].

In this study, we included baseline demographic, preoperative, and operative factors (total 25 features), to predict postoperative infections and identify associated important features. Demographic features included age, gender, ethnicity, marital status, and insurance status. Preoperative features encompassed health status indicators such as obesity, depression, fluid and electrolyte disorders, coagulopathy, and liver disease, as well as behavioral factors like alcohol or drug abuse history, and hospital assessment details including admission type and location. The operative feature pertained to the surgery type, such as cardiac or neurologic surgery. Among the surgery patients, the average age was 64.2 years. The gender distribution was predominantly male (59.63%), with females accounting for 40.37%. Medicare/Government insurance covered most patients (54.68%), followed by Self-Pay/Private (38.62%) and Medicaid (6.70%). Ethnically, the cohort was predominantly White (73.39%), with 13.23% of unknown ethnicity, 5.6% Black/African, 4.84% other ethnicities, and 2.94% Hispanic/Latino. For marital status, 54.81% of patients were married, 20.35% were single, 12.73% were widowed, 7.22% were divorced or separated, and 4.89% fell into other categories. The most common types of surgery were

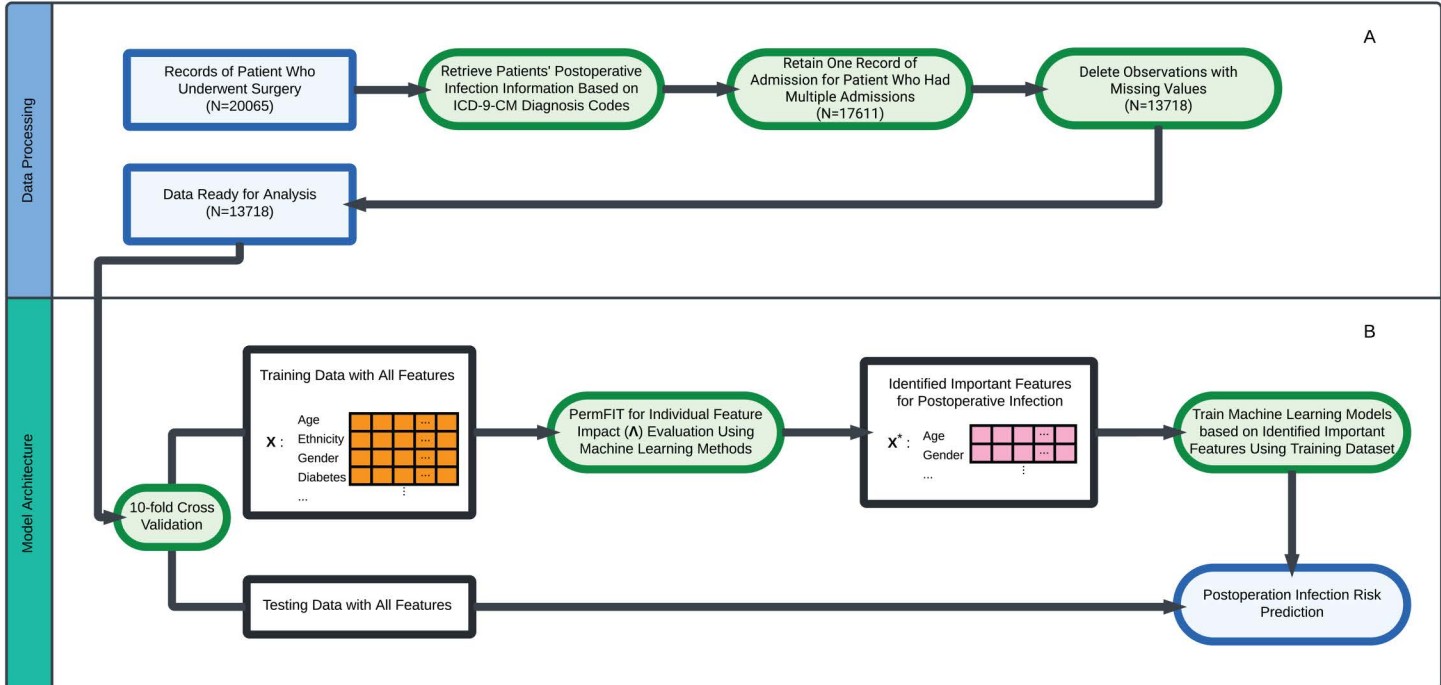

**Fig 1. Data Processing and Model Architecture.**

cardiac (40.45%), general (22.70%), neurologic (19.97%), thoracic (5.68%), musculoskeletal (4.7%), circulatory (5.45%), and plastic (1.05%). Comorbidities varied, with notable percentages for hypertension (58.28%), diabetes (26.54%), and congestive heart failure (19.9%).

**Machine Learning Models for Postoperative Infection Risk Prediction** To predict the postoperative infection risk, denoted as $Z$ (where 1 represents positive and 0 represents negative), based on a set of clinical features $\mathbf{X} = (X_1, \ldots, X_p)$ (e.g., age, gender, obesity, surgery type, etc.), we need to assess the conditional probability of being in a positive postoperative infection state given the input predicting features $\mathbf{X}$. This is represented as $\pi(\mathbf{X}) = E(Z|\mathbf{X}) = Pr(Z = 1|\mathbf{X})$.

Traditionally, logistic regression models are used for this purpose, assuming that these features impact the probability of a positive postoperative infection in a generalized linear additive manner. However, this assumption rarely holds in practice and is difficult to verify. To mitigate this restrictive assumption, machine learning methods are often employed. We compare commonly used machine learning models—SVM, XGBoost, RF, and DNN—for their performance in classifying postoperative infection status. Unlike conventional DNN methods, we adopt a stable DNN procedure that utilizes bootstrap aggregating and scoring algorithms to filter out unstable bootstrapped DNN models, significantly improving prediction precision [31]. Though machine learning methods can flexibly model the complex associations between risk features and postoperative infection risk status without the need to specify the association explicitly, they are black-box machines, lacking the capability to evaluate each individual feature's impact and hindering these methods from being adopted in clinical practice. To address this challenge for postoperative infection risk prediction using machine learning models, we adopt a permutation feature importance test as described below.

**PermFIT for Interpretable Postoperative Infection Risk Prediction** We adopt the following permutation-based feature importance test procedure for various machine learning models to afford robust statistical inference with a principled statistical inference and identify the statistically important clinical features associated with postoperative infection. The

feature importance score $\Lambda_j$ of feature $X_j$ (i.e., the $j^{th}$ feature in $\mathbf{X}$ (j = 1, ..., p)) is defined as the expected squared difference between $\pi(\mathbf{X})$ and $\pi(\mathbf{X}^{(j)})$ where $\mathbf{X}^{(j)} = (X_1, ..., X_{j-1}, X_{j'}, X_{j+1}, ..., X_p)$ is $\mathbf{X}$ with its $j^{th}$ feature replaced by $X_{j'}$, a random permutation of the elements of $X_j$. The importance score $\Lambda_j$ can be re-expressed as $\Lambda_j = E_{X, X_{j'}}[\pi(\mathbf{X}) - \pi(\mathbf{X}^{(j)})]^2$, which is zero only when $\pi(\mathbf{X}) \equiv \pi(\mathbf{X}^{(j)})$, implying no contribution of $\mathbf{X}^{(j)}$ on $\pi(\mathbf{X})$ conditional on the other features. The larger the impact of $\mathbf{X}^{(j)}$ on $\pi(\mathbf{X})$, the larger $\Lambda_j$ is expected to be. Furthermore, $\Lambda_j$ can be estimated as the empirical per-

mutation importance score $\Lambda_j^{(P)} = \frac{1}{n}\sum_{i=1}^{n}\Lambda_{ij}^{(P)}$ where $\Lambda_{ij}^{(P)} = Z_i log\left(\frac{\hat{\pi}(\mathbf{X}_{i.})}{\hat{\pi}(\mathbf{X}_{i.}^{(j)})}\right) + (1 - Z_i) log\left(\frac{1 - \hat{\pi}(\mathbf{X}_{i.})}{1 - \hat{\pi}(\mathbf{X}_{i.}^{(j)})}\right)$ with $\mathbf{X}_{i.} = (X_{i1}, \ldots, X_{ip})$

and $\mathbf{X}_{i.}^{(j)} = (X_{i1}, \ldots, X_{i,j-1}, X_{s_i,j}, X_{i,j+1}, \ldots, X_{ip})$. The estimate of $\pi(\cdot)$, i.e., $\hat{\pi}(\cdot)$, can be obtained using SVM, XGBoost, RF, and DNN. Particularly, the DNN method we use is the stable DNN by Mi et al [31]. We then estimate $\Lambda_j^P$ as $\hat{\Lambda}_j^{(P)}$, and the variance estimate of $\hat{\Lambda}_j^{(P)}$ is given as $\hat{Var}\left[\hat{\Lambda}_j^{(P)}\right]$. Based on it, we construct the test statistic for importance hypothesis test of

feature Xj as: $\lambda = \frac{\hat{\Lambda}_j^{(P)}}{\sqrt{\hat{Var}\left[\hat{\Lambda}_j^{(P)}\right]}}$. The details of overall our proposed modeling architecture are depicted in Fig 1B.

**Statistical Analysis** To comprehensively evaluate model prediction performance, we adopted a 10-fold cross-validation strategy, as illustrated in Fig 1B. Each fold alternately served as the testing set (10% of samples) and the training set (90% of samples). The training set was used for feature importance identification and predictive model retraining based on the identified important features for each machine learning model. To ensure fair comparison, we kept identical cross-validation splits for all models. We applied this methodology across four machine learning models: SVM, XGBoost, RF, and DNN, utilizing all 25 features as input. Additionally, the significant features for each machine learning model were determined at a significance level of 0.05 within the training dataset. Each model's performance was evaluated based on the testing dataset using metrics including accuracy, area under the ROC curve (AUC), sensitivity, specificity, and their respective 95% confidence intervals. Furthermore, because postoperative infection occurs in approximately 25% of patients in our dataset, the outcome distribution is naturally imbalanced. Rather than applying oversampling techniques such as SMOTE [43], which may alter the joint distribution between predictors and outcomes and potentially distort the underlying association structure, we preserved the original data distribution and evaluated model performance using precision-recall area under the curve (PR-AUC), which is more informative for imbalanced classification problems.

In our DNN method, we employed an architecture with four hidden layers with decreasing node counts (50, 40, 30, 20). The risk function was optimized using a mini-batch stochastic gradient descent algorithm with adaptive learning rate adjustment. The models underwent training for 1000 epochs with a mini-batch size of 30. The number of optimal DNN models to be kept in the final ensemble was determined by minimizing the loss on training samples. For the random forest model, we utilized an ensemble of 1000 decision trees, each constructed with a node size of 3. We employed the XGBoost R package and determined tuning parameters through cross-validation, optimizing key parameters including a learning rate of 0.3 and a maximum tree depth of 5, with up to 5 boosting iterations. The SVM model underwent tuning, involving variation of the gamma and cost parameters in logarithmic steps. The best combination was determined using cross-validation. We ensured that the data used for hyper-parameter tuning and performance evaluation were distinct. The data used for model performance evaluation were never exposed to the model during training in each round of cross-validation, preventing data leakage and reducing overfitting.

## Results

Under the permutation feature importance test framework, we evaluated the importance of each of the 25 features using the aforementioned feature importance statistics for each of the four commonly used machine learning models: DNN, RF, SVM, and XGBoost. With the identified significant features associated with postoperative infection for each machine learning model, only these identified important features were incorporated into the corresponding machine learning model to predict postoperative infection risk, referred to as the interpretable model and denoted as DNN, RF, SVM,

and XGBoost, respectively. The analysis results are presented in Table 1, and the predicted AUC and PR-AUC values from 10-fold cross-validation using the identified important features for each machine learning model are depicted in Fig 2.

The results presented in Table 1 demonstrate that the stable DNN method outperforms all other machine learning methods in predicting postoperative infection using the identified important features, with mean (95% CI) values of accuracy, AUC, PR-AUC, sensitivity, and specificity as 0.817 (0.811, 0.822), 0.799 (0.790, 0.808), 0.654 (0.642, 0.666), 0.324 (0.311, 0.336), and 0.983 (0.980, 0.986), respectively. These findings clearly demonstrate the superiority of the introduced stable DNN method for postoperative infection prediction and its effectiveness in identifying important associated features. The statistical inference of each individual feature's impact on postoperative infection not only enhances our understanding of the postoperative infection mechanism under complex association settings but also potentially improves prediction performance by effectively excluding many noise features.

**Important Features Associated with Postoperative Infection Risk** Under the permutation feature importance test framework, we determined the important features associated with postoperative infection for each machine learning model across each fold of 10-fold cross-validation. We plotted the heatmap of frequency for identified important features of each machine learning method at the significance level of 0.05 in Fig 3.

Table 1. Model Performance Comparison on Postoperative Infection Prediction.

| Metric | DNN | RF | SVM | XGBoost |
|---|---|---|---|---|
| Accuracy (95% CI) | 0.817 (0.811, 0.822) | 0.815 (0.810, 0.820) | 0.815 (0.809, 0.821) | 0.816 (0.810, 0.822) |
| AUC (95% CI) | 0.799 (0.790, 0.808) | 0.665 (0.654, 0.676) | 0.670 (0.647, 0.692) | 0.787 (0.778, 0.796) |
| PR-AUC (95% CI) | 0.654 (0.642, 0.666) | 0.651 (0.640, 0.661) | 0.568 (0.550, 0.586) | 0.648 (0.636, 0.659) |
| Sensitivity (95% CI) | 0.324 (0.311, 0.336) | 0.314 (0.303, 0.324) | 0.325 (0.312, 0.338) | 0.326 (0.314, 0.339) |
| Specificity (95% CI) | 0.983 (0.980, 0.986) | 0.984 (0.982, 0.987) | 0.981 (0.977, 0.985) | 0.982 (0.978, 0.985) |
| F1 Score (95% CI) | 0.471 (0.459, 0.483) | 0.461 (0.450, 0.472) | 0.471 (0.458, 0.484) | 0.472 (0.459, 0.485) |

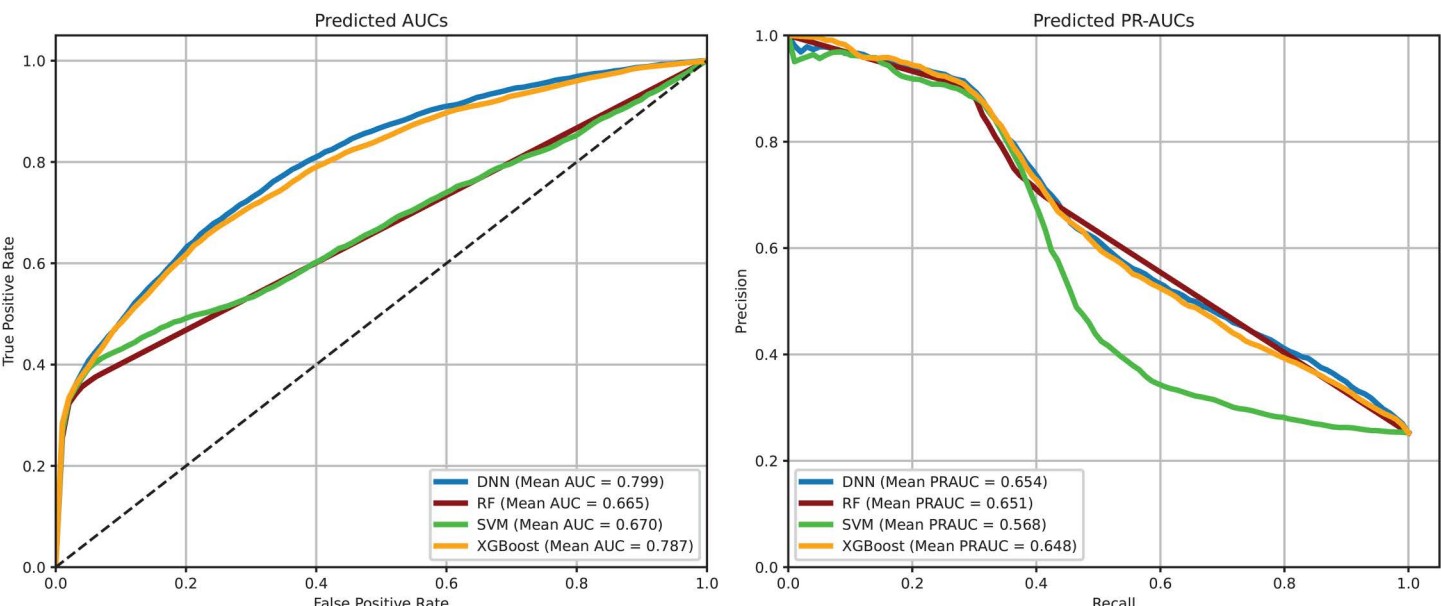

Fig 2. Average AUCs and PR-AUCs for Machine Learning Models.

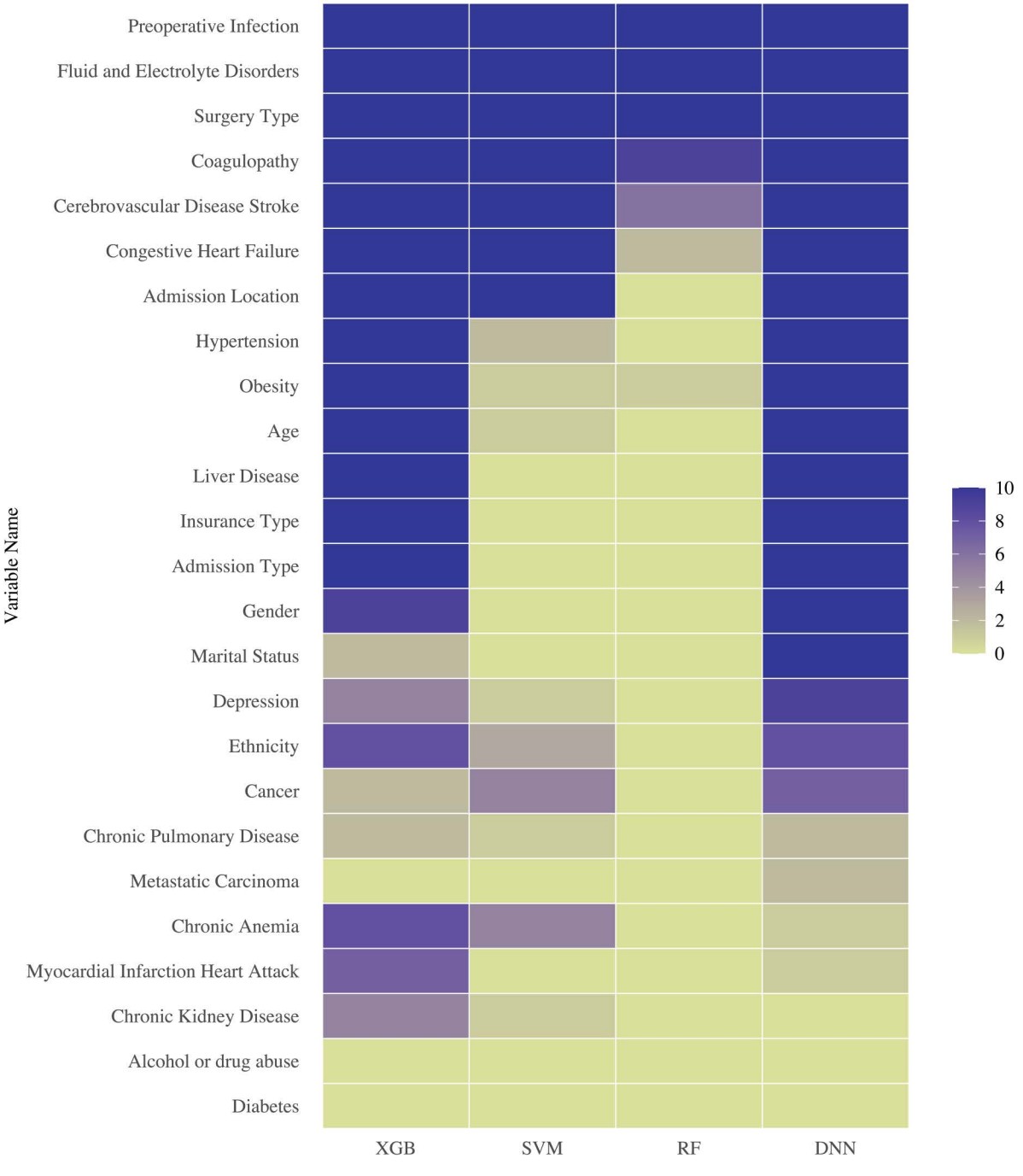

**Fig 3. Heatmap of Frequency of Identified Significant Variable.**

It is evident that preoperative infection, fluid and electrolyte disorders, and surgery type are consistently identified as significant factors for postoperative infection across all 10 folds by all four methods. Additionally, coagulopathy, admission location, admission type, cerebrovascular disease, congestive heart failure, hypertension, obesity, liver disease, age, insurance type, gender, and marital status were all identified as significant factors across all 10 folds by the DNN model. Among these significant findings, many are consistent with earlier study findings, and some are novel. For instance, preoperative infection, age, congestive heart failure, hypertension, obesity, and depression have been shown to significantly impact postoperative infection [44–46]. While earlier studies did not indicate gender as a significant factor for postoperative infection [45], our study identifies gender as an important factor significantly influencing the development of postoperative infection. On the other hand, both RF and SVM methods failed to identify important features such as hypertension, obesity, age, liver disease, etc, which have been verified in earlier clinical studies [47–50]. Though XGBoost can achieve almost identical prediction performance to DNN using the identified important features, XGBoost failed to identify some important features that have been clinically verified such as depression [49]. All these clearly suggest that DNN coupled with PermFIT is a promising tool to identify important features associated with postoperative infection.

## Discussion

Developing robust models for predicting postoperative infections in ICU surgical patients and identifying critical factors associated with these infections is crucial for effective postoperative management and improved patient outcomes. In this study, we employed several machine learning models, including SVM, XGBoost, RF, and DNN, to predict postoperative infection risk using large-scale electronic health records. Our findings indicate that the DNN model slightly outperformed the other methods in terms of prediction accuracy, sensitivity, specificity, AUC, and PR-AUC, suggesting its superiority in capturing complex associations between risk factors and postoperative infection. Most importantly, while most of the important features identified by the DNN model have been verified in previous clinical studies, other machine learning methods often failed to capture these important features.

Under the permutation feature importance test framework, the stable DNN model identified 15 significant risk factors associated with postoperative infection. These factors, derived from demographic variables, medical history, and surgery type, provided valuable insights into the determinants of postoperative infection mechanisms. Leveraging these identified features, the DNN model achieved the best performance across all evaluation metrics, indicating its ability to effectively utilize relevant clinical data for precise risk estimation. Our study underscores the significance of machine learning techniques in postoperative infection prediction and highlights the utility of permutation feature importance testing in identifying clinically relevant risk factors. By integrating these advanced analytical approaches into clinical practice, healthcare providers can better identify patients at higher risk of postoperative infection, enabling timely interventions and improved outcomes [51]. Additionally, the comprehensive evaluation of machine learning models and feature importance testing conducted in this study provides a robust framework for future research aimed at refining postoperative infection prediction models and identifying additional risk factors contributing to surgical complications.

A notable limitation of this study is that the MIMIC-III data did not include important patient factors such as smoking status [52,53] and nutritional status [54,55], which are known to influence the development of postoperative infections, yet they are not available in the data set analyzed in this study. The omission of these variables could limit the comprehensiveness of our predictive model and affect the accuracy of risk prediction. Integrating these additional factors and developing more comprehensive models will enhance the accuracy and applicability of postoperative infection risk predictions, ultimately improving patient care and outcomes. Also, it should be noted that the feature importance analysis should be interpreted primarily as an exploratory variable identification procedure within a predictive modeling framework, rather than strict confirmatory hypothesis testing.

## Conclusions

Our study demonstrates the effectiveness of the stable DNN model in conjunction with the permutation feature importance test framework in evaluating each input factor's impact on predicting postoperative infection risk. By identifying key risk factors and employing advanced computational methodologies, our research contributes to a deeper understanding of postoperative infection mechanisms, improving the quality of postoperative care and optimizing outcomes for ICU patients. Further validation and integration of these predictive models into clinical practice hold promise for advancing personalized postoperative care and management for ICU patients.

## Acknowledgments

**We would like to thank the academic editor and two reviewers for the constructive comments and suggestions.**

## Author contributions

**Conceptualization:** Fei Zou, Baiming Zou.

**Data curation:** Xiaoyu Wu.

**Formal analysis:** Xiaoyu Wu.

**Investigation:** Isaac Luria, Patrick Tighe.

**Methodology:** Fei Zou, Baiming Zou.

**Supervision:** Fei Zou, Baiming Zou.

**Validation:** Xiaoyu Wu.

**Writing – original draft:** Xiaoyu Wu, Baiming Zou.

**Writing – review & editing:** Xiaoyu Wu, Isaac Luria, Meisheng Xiao, Patrick Tighe, Fei Zou, Baiming Zou.

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
