## [Decision Letter · Decision Letter 0]

6 Jul 2025

Dear Dr. Zou,

Thank you for submitting your manuscript to PLOS ONE. After careful consideration, we feel that it has merit but does not fully meet PLOS ONE’s publication criteria as it currently stands. Therefore, we invite you to submit a revised version of the manuscript that addresses the points raised during the review process.

We look forward to receiving your revised manuscript.

Kind regards,

Young-Seob Jeong

Academic Editor

PLOS ONE

Journal Requirements:

https://journals.plos.org/plosone/s/file?id=wjVg/PLOSOne_formatting_sample_main_body.pdf andandandand

Precision Opioid Prescription in ICU Surgery: Insights from an Interpretable Deep Learning Framework - https://doi.org/10.29011/2575-9760.11189

(Among others)

In your revision ensure you cite all your sources (including your own works), and quote or rephrase any duplicated text outside the methods section. Further consideration is dependent on these concerns being addressed.

“This study was partially supported by NIH (National Institutes of Health) R56 (1R56LM013784) and R01 (R01LM014407 and 1R01HL173044) grants.”

“This study was partially supported by NIH (National Institutes of Health) R56 (1R56LM013784) and R01 (R01LM014407 and 1R01HL173044) grants.”

“This study was partially supported by NIH (National Institutes of Health) R56

(1R56LM013784) and R01 (R01LM014407 and 1R01HL173044) grants.”

“This study was partially supported by NIH (National Institutes of Health) R56 (1R56LM013784) and R01 (R01LM014407 and 1R01HL173044) grants.”

6. We note that you have indicated that there are restrictions to data sharing for this study. For studies involving human research participant data or other sensitive data, we encourage authors to share de-identified or anonymized data. However, when data cannot be publicly shared for ethical reasons, we allow authors to make their data sets available upon request. For information on unacceptable data access restrictions, please see http://journals.plos.org/plosone/s/data-availability#loc-unacceptable-data-access-restrictions.

Reviewers' comments:

Reviewer's Responses to Questions

**Comments to the Author**

1. Is the manuscript technically sound, and do the data support the conclusions?

Reviewer #1: Yes

Reviewer #2: Yes

2. Has the statistical analysis been performed appropriately and rigorously?

Reviewer #1: No

Reviewer #2: Yes

3. Have the authors made all data underlying the findings in their manuscript fully available?

Reviewer #1: Yes

Reviewer #2: Yes

4. Is the manuscript presented in an intelligible fashion and written in standard English?

Reviewer #1: Yes

Reviewer #2: Yes

Reviewer #1: This manuscript presents a potentially valuable contribution to the field of clinical predictive modeling by focusing on the integration of model interpretability and stability into deep learning-based risk prediction. The problem is well-motivated, and the use of permutation-based feature importance as an interpretability tool is relevant. The use of ensemble DNNs trained with a filtering mechanism to improve model stability is also a promising direction.

Some terminology is introduced without precise definition (e.g., “stable DNN model”) or where claims are made without supporting citations (e.g., about the limitations of interpretability in existing ML models used in ICU prediction).

However, the current version of the manuscript overstates the practical utility of the approach given the results. While the DNN ensemble performs marginally better than existing models, the improvement in AUC and PR-AUC is incremental, and the low sensitivity across all methods suggests that the model may not be suitable for critical clinical use without further refinement. The lack of external validation, absence of confidence intervals or statistical comparisons of feature importance, and limited discussion of generalisability or deployment constraints also weaken the broader applicability of the findings.

The paper would be strengthened by the following:

- More thorough statistical evaluation of model outputs, particularly in terms of calibration and variance.

- Additional detail on how the DNN ensemble was constructed, filtered, and how unstable models were defined or excluded.

- Clarification of feature selection strategy and justification for excluded clinically relevant features such as smoking status or nutritional indicators.

- Deeper comparison of the PermFIT approach to existing interpretable ML methods (e.g., SHAP, LIME).

In its current form, the paper presents an interesting methodological direction but needs additional clarity, statistical rigour, and critical reflection before it can fully support the conclusions it puts forward. If these areas are addressed, the manuscript has the potential to contribute meaningfully to the literature on interpretable machine learning in critical care settings.

Reviewer #2: As a reviewer evaluating this manuscript titled “An Interpretable Deep Learning Framework for Predictive Modeling of Postoperative Infections in ICU Patients”, submitted to PLOS ONE, I provide the following critical scientific assessment along with specific questions and suggestions to strengthen its rigor, clarity, and reproducibility.

This manuscript addresses an important clinical challenge—predicting postoperative infections in ICU patients—by integrating deep learning with a novel permutation-based feature importance test (PermFIT) to improve both prediction and interpretability. The study uses a large, publicly available dataset (MIMIC-III), implements multiple machine learning models, and performs systematic comparisons. The results indicate that the proposed stable DNN outperforms other models in predictive metrics and interprets key risk factors effectively.

The study is well-motivated and relevant to PLOS ONE's scope, but several critical issues require clarification or improvement before the manuscript can be considered for publication.

Major Scientific Comments & Questions

1. While the PermFIT approach is emphasized as novel, the authors cite their own previous work (Mi et al., 2021; Ref [41]) that also introduced a similar DNN model. How does this work advance that prior publication?

2. Despite high AUC and specificity, sensitivity is very low (~32%), meaning a large proportion of infected patients would be missed. Is this acceptable for a clinical prediction model?

3. The authors mention 25 input features but do not explain why these were chosen or whether other known infection risk factors (e.g., surgical duration, antibiotic use) were considered.

4. How were categorical variables encoded? Were patients with multiple ICU admissions properly deduplicated? Could leakage from label-informed preprocessing have inflated results?

5. The study uses MIMIC-III exclusively. Have the authors tested the model on MIMIC-IV or another external dataset?

6. How were p-values corrected for multiple hypothesis testing (e.g., Bonferroni or FDR)? With 25 features, there’s a high risk of Type I error.

7. The authors report high specificity but low sensitivity. Were additional metrics such as F1-score, Brier score, or calibration curves considered?

Minor Issues and Suggestions

• Clarify acronym definitions at first use (e.g., EHR, DNN).

• Move all figures and tables closer to their first mention for better readability.

• Ensure all references (e.g., Ref [39] for MIMIC-III) link to publicly accessible data sources, and include IRB or ethical approval statements if applicable.

• Typo: “healthcare-associated infections” is occasionally misspelled as “healthcare-associate infections.”

Summary Recommendation

This study addresses an important topic with promising methodology, but it requires substantial revisions before it meets publication standards. The main concerns are:

• Sensitivity and clinical applicability of the model

• Transparency about methodological choices

• Clarification of novelty

• Lack of external validation or calibration

If these concerns are properly addressed, the manuscript would be a strong candidate for publication.

.

Reviewer #1: No

Reviewer #2: **Yes:** Randa A AbdelnaserRanda A AbdelnaserRanda A AbdelnaserRanda A Abdelnaser

---

## [Author Response · Author response to Decision Letter 1]

22 Jul 2025

Please see the uploaded Response.pdf file.

---

## [Decision Letter · Decision Letter 1]

9 Mar 2026

Dear Dr. Zou,

Thank you for submitting your manuscript to PLOS ONE. After careful consideration, we feel that it has merit but does not fully meet PLOS ONE’s publication criteria as it currently stands. Therefore, we invite you to submit a revised version of the manuscript that addresses the points raised during the review process.

We look forward to receiving your revised manuscript.

Kind regards,

Young-Seob Jeong

Academic Editor

PLOS One

Journal Requirements:

Reviewers' comments:

Reviewer's Responses to Questions

**Comments to the Author**

Reviewer #2: All comments have been addressed

Reviewer #3: (No Response)

2. Is the manuscript technically sound, and do the data support the conclusions?

Reviewer #2: Yes

Reviewer #3: (No Response)

3. Has the statistical analysis been performed appropriately and rigorously?

Reviewer #2: Yes

Reviewer #3: (No Response)

4. Have the authors made all data underlying the findings in their manuscript fully available?

Reviewer #2: Yes

Reviewer #3: (No Response)

5. Is the manuscript presented in an intelligible fashion and written in standard English?

Reviewer #2: Yes

Reviewer #3: (No Response)

Reviewer #2: Reviewer #2 – Final Comments

The authors have adequately addressed all major and minor concerns raised in the initial review. They have clarified the novelty of applying their previously developed PermFIT and DNN methods to identify key features associated with postoperative infection, and they have acknowledged the limitations regarding sensitivity, feature availability, and external validation. The revised manuscript is significantly improved in clarity and presentation. I find the work scientifically sound, relevant, and suitable for publication in PLOS ONE.

Recommendation: Accept

Reviewer #3: The proposed technique exhibits effective results. My concerns are as follows:

1. No correction for multiple testing (e.g., FDR or Bonferroni) in feature importance p-values increases risk of false positives among the 25 features.

2. Marginal performance gains of DNN over XGBoost (e.g., AUC 0.799 vs. 0.787) do not strongly justify its superiority, especially given computational complexity.

3. Some references are worth mentioning: DOI: 10.62762/TETAI.2024.532253; DOI: 10.62762/BISH.2025.457428; DOI: 10.62762/BISH.2025.352565

4. The reviewer is wondering how about the computational complexity of the proposed method?

5. Imbalanced outcome (inferred ~25% infection rate) not explicitly addressed; techniques like SMOTE or weighted loss could improve sensitivity.

.

Reviewer #2: **Yes:** Randa A AbdelnaserRanda A AbdelnaserRanda A AbdelnaserRanda A Abdelnaser

Reviewer #3: No

---

## [Author Response · Author response to Decision Letter 2]

24 Mar 2026

Please see attached PDF file for point-by-point response to reviewer comments.

---

## [Editor Report · Decision Letter 2]

25 Mar 2026

An Interpretable Deep Learning Framework for Predictive Modeling of Postoperative Infections in ICU Patients

PONE-D-25-12411R2

Dear Dr. Zou,

We’re pleased to inform you that your manuscript has been judged scientifically suitable for publication and will be formally accepted for publication once it meets all outstanding technical requirements.

Kind regards,

Young-Seob Jeong

Academic Editor

PLOS One

Additional Editor Comments (optional):

The authors have provided thorough responses to all of the reviewers’ comments, and the quality of the manuscript is sufficient for publication in this journal.